# On Weather Data-Based Prediction of Gamma Exposure Rates Using Gradient Boosting Learning for Environmental Radiation Monitoring

**DOI:** 10.3390/s22187062

**Published:** 2022-09-18

**Authors:** Changhyun Cho, Kihyeon Kwon, Chase Wu

**Affiliations:** 1Division of Electronic, Information and Communication Engineering, Kangwon National University, Samcheok 25913, Kangwondo, Korea; 2Department of Data Science, New Jersey Institute of Technology, Newark, NJ 07102, USA

**Keywords:** time-series data analysis, gamma exposure rate, data preprocessing, gradient boosting, LSTM, LightGBM

## Abstract

Gamma radiation has been classified by the International Agency for Research on Cancer (IARC) as a carcinogenic agent with sufficient evidence in humans. Previous studies show that some weather data are cross-correlated with gamma exposure rates; hence, we hypothesize that the gamma exposure rate could be predicted with certain weather data. In this study, we collected various weather and radiation data from an automatic weather system (AWS) and environmental radiation monitoring system (ERMS) during a specific period and trained and tested two time-series learning algorithms—namely, long short-term memory (LSTM) and light gradient boosting machine (LightGBM)—with two preprocessing methods, namely, standardization and normalization. The experimental results illustrate that standardization is superior to normalization for data preprocessing with smaller deviations, and LightGBM outperforms LSTM in terms of prediction accuracy and running time. The prediction capability of LightGBM makes it possible to determine whether the increase in the gamma exposure rate is caused by a change in the weather or an actual gamma ray for environmental radiation monitoring.

## 1. Introduction

Gamma ray or gamma radiation is an electromagnetic wave generated when an atomic nucleus in an excited energy state moves to a lower state or ground state or when a particle is annihilated [1]. When a human body or living organism is irradiated with gamma rays for a long time, cells might be destroyed and DNA chains might be broken. Therefore, it is designated as a Group 1 carcinogen by the World Health Organization’s (WHO) International Agency for Research on Cancer (IARC) [2].

High-pressure ion chambers (HPICs) are gas-filled detectors which respond to gamma energies and have been deployed for environmental and area monitoring. Gamma radiation causes a current to flow in an ion chamber detector. The magnitude of this electric current is proportional to the exposure rate [3,4].

In view of the threat of gamma rays to human health, various studies have been conducted in the literature [5,6]. Many previous efforts were focused on the health effects of gamma rays on artificial nuclides, but, more recently, researchers have started to investigate the health effects of radon, which is a representative natural gamma ray [7]. As gamma rays are dangerous in both artificial and natural materials, spatial gamma dose rates are analyzed in various places [8,9]. For example, autonomous vehicles are equipped with devices that enable them to observe their environments and make decisions in real time [10].

Previous studies have revealed that there exists a certain correlation between spatial gamma dose rate and rainfall data [11,12]. Specifically, preliminary results show that the correlation is highest with a time scale of one day and decreases as the time scale increases to one month or one year.

Considering such a correlation between the gamma exposure rate and the weather data, and if we can predict the change in the gamma exposure rate with the weather data, it could help us to understand and determine the actual cause of the increase in the gamma exposure rate.

In this study, we collected various weather and radiation data from the automatic weather system (AWS) and the environmental radiation monitoring system (ERMS) during a specific period and trained and tested two time-series learning algorithms—namely, long short-term memory (LSTM) and light gradient boosting machine (LightGBM)—with two preprocessing methods, namely, standardization and normalization. The experimental results illustrate that standardization is superior to normalization for data preprocessing with smaller deviations, and LightGBM outperforms LSTM in terms of prediction accuracy and running time. The prediction capability of LightGBM makes it possible to determine whether the increase in the gamma exposure rate is caused by a change in the weather or an actual gamma ray from radioactive materials.

## 2. Experimental Dataset

To support our research, we collected weather data from the automatic weather system (AWS) and radiation data from the high-pressure ion chamber (HPIC) of the environmental radiation monitoring system (ERMS) located in Uljin-gun, Korea. As the correlation between the weather observation data and gamma exposure rate decreases as the year progresses, we focused on the weather and radiation measurements during a period from 0:00 on 1 July 2020 to 0:00 on 1 November 2020, during which the weather had undergone severe changes. Specifically, during this period of four months, we acquired the 5-min average of gamma exposure rates, totaling 35,424 data points, and the 5-min average of weather measurements, including ground temperature, ground humidity, rainfall, atmospheric pressure, temperature at a 10 m tower, wind direction at a 10 m tower, wind speed at a 10 m tower, maximum wind speed at a 10 m tower, temperature at a 58 m tower, wind direction at a 58 m tower, wind speed at a 58 m tower, and maximum wind speed at a 58 m tower, as shown in Table 1, totalling 425,052 data points (35,421 × 12).

## 3. Proposed Research Methods

### 3.1. Data Preprocessing

There were three missing 5-min averages of meteorological measurements. For the continuity of the dataset, these missing measurements were filled in with the same value as the preceding 5-min average. Furthermore, to unify various types of data in different engineering units, we employed two preprocessing methods, namely, normalization and standardization, and compared their effects on the performance.

Normalization

Min–max Scaling was used for normalization, which adjusts the data so that all values are between 0 and 1. This can be obtained by the following formula:(1)$${x_{i}}^{\prime }=\frac{x_{i}-x_{min}}{x_{max}-x_{min}}$$
where $x_{max}$ is the maximum value of the data, and $x_{min}\;$ is the minimum value of the data.

2.Standardization

Standardization is the adjustment of data so that it has the properties of a normal distribution (μ=0, σ=1). This can be obtained by the following formula:(2)$${x_{i}}^{\prime }=\frac{x_{i}-\mu }{\sigma }$$
where μ is the mean value of all data, and σ is the standard deviation of all data.

3.Data split

The dataset was divided into three parts: 80% of the total data was used for model training, 10% as a validation set for model optimization, and 10% as a test set to test the prediction results of the model.

### 3.2. Learning Algorithms

Considering the time-serial nature of the weather and radiation dataset, we chose long short-term memory (LSTM), a variant of a recurrent neural network (RNN), as a model for prediction. Since there exists a certain level of variation in such time-series data collected from real environments, we investigated the application of an ensemble learning framework, light gradient boosting machine (LightGBM). Both LSTM and LightGBM were trained with the data from the AWS and ERMS.

4.LSTM

The long short-term memory (LSTM) network is an improved model of a recurrent neural network (RNN) that specifically addresses the long-term dependency problem of an RNN. The LSTM network is intended to classify time-series data or learn long-term dependencies between data. Generally, the LSTM network is divided into an input layer and an LSTM layer. The input layer receives time-series data, and the LSTM layer learns long-term dependencies between time-series data [13,14].

5.LightGBM

Gradient boosting machine (GBM) combines weak learners to form strong learners, using gradient descent to assign weights. The decision tree used for GBM is expanded in a level-wise manner, and multiple decision trees are combined to predict the result [15].

Although GBM has shown good learning results, it suffers from a problem of low efficiency in processing a large amount of data. Light gradient boosting machine (LightGBM) is a model belonging to a boosting series among ensemble learning models. Boosting is an algorithm that creates several weak models using gradient descent, trains them sequentially, and builds a model that is weighted according to the performance of the previous model. Unlike other models, LightGBM uses leaf-wise partitioning to perform learning in a way that reduces the loss in model training more than the level-wise partitioning method and has the advantage of taking less training time. The difference between leaf-wise and level-wise is illustrated in Figure 1 [16].

## 4. Implementation and Performance Evaluation

### 4.1. Data Analysis

We implemented the chosen learning methods using Python version 3.9.7, TensorFlow version 2.7.0, and LightGBM version 3.3.2. The technical solution integrates various components such as data preprocessing, analysis, and learning, and is executed on a Jupyter notebook with Anaconda 3 2021.11. The hardware specifications used in the experiments are provided in Table 2.

Since our goal was to predict the gamma exposure rate through the weather data, we needed to investigate the correlation between the weather data and the gamma exposure rate. Since the correlation between rainfall and gamma exposure is known, we conducted data analysis to find out if there existed significant correlations in other weather data. Towards this goal, we first performed a visual examination of the data by creating the scatter plot of the measurements in Figure 2, which exhibits certain patterns between the variables. We further computed the correlation coefficient and the *p*-value as shown in Table 3. Note that for the correlation coefficient, the closer to 1, the higher the positive correlation, and the closer to −1, the higher the negative correlation. If it is 0, there is no correlation at all [17]. If the *p*-value is less than 0.05, it indicates that there is a significant relationship between the control and response variables [18].

The correlation analysis results show that there does not exist a high correlatioin between the weather data and the gamma exposure rate, and some variables show a low correlation coefficient of less than 0.1. We further computed the *p*-value to check if the data were meaningful as the learning data, and it was confirmed that all were less than 0.05. Note that the data were preprocessed by two different methods, namely, standardization and normalization, before training the models.

### 4.2. Learning Results

#### 4.2.1. LSTM-Based Learning Model

We designed an LSTM model with a hidden layer of 16 nodes. There was no significant change in the learning result even when the batch size was larger than 16. The number of trainings was 200 and the batch size was set to 16. We used mean square error (MSE) as the loss function, Adam as the optimizer, and sigmoid as the activation function. When ReLU is chosen as the activation function, GPU learning based on a CUDA deep neural network (cuDNN) is not available, and the learning result does not change significantly from that of sigmoid. EarlyStopping was set to prevent overfitting and learning was stopped when the loss function did not improve more than five times. For the comparison of the learning results, we considered mean square error (MSE) and root mean square error (RMSE) as the metrics, which can be computed as follows:(3)$$MSE=\frac{1}{n}\overset{n}{\underset{i=1}{\sum }}{\left(y_{i}-{\hat{y}}_{i}\right)}^{2}$$
(4)$$RMSE=\sqrt{MSE}$$
where $y_{i}$ is the estimate value of the model, and ${\hat{y}}_{i}$ is the target value of the model.

As shown in Table 4, the LSTM learning model preprocessed with standardization achieves a RMSE of 0.7729, and the LSTM learning model preprocessed with normalization achieves a RMSE of 0.0433. These results indicate that normalization preprocessing seems to be more advantageous in reducing the learning errors.

To understand how well the learned model would behave in prediction, we plotted the regression curve between the learned values and the actual values for each model, as shown in Figure 3. These regression curves show that there is a notable discrepancy in both of the learning models. However, considering that LSTM with standardization preprocessing yields a larger slope of the regression curve, it is our conjecture that the standardization preprocessing method would help LSTM achieve a better prediction accuracy.

We tested the trained LSTM model with different preprocessing methods and plotted the corresponding prediction curves in Figure 4. From these prediction curves, we observe that the LSTM model with standardization preprocessing achieves a satisfactory prediction performance, while the LSTM model with normalization preprocessing does not perform well. This is consistent with our conjecture. These results also indicate that the preprocessing method has a significant impact on the prediction performance.

#### 4.2.2. LightGBM-Based Learning Model

We designed a LightGBM model in which the learning rate is set to 0.01, max depth is set to 16, boosting is based on GBDT, the number of leaves is set to 144, objective function uses regression, feature fraction is set to 0.9, bagging fraction is set to 0.7, bagging frequency is set to 5, seed is set to 2018, and the metric uses the area under the curve (AUC). To prevent overfitting, the training process was stopped early when the optimal AUC was calculated over 1000 rounds. However, if the learning process ends before reaching 1000 rounds, it does not yield an accurate learning result. We also observe that the changes in other parameters do not significantly affect the learning results. Similarly, to compare the learning results, we considered mean square error (MSE) and root mean square error (RMSE) as the metrics.

As shown in Table 5, the LightGBM learning model preprocessed by standardization achieves a RMSE of 0.38441478, and the LightGBM learning model preprocessed by normalization achieves a RMSE of 0.0337. These results indicate that normalization preprocessing seems to be more advantageous in reducing the learning errors. Moreover, we plotted the regression curves of the LightGBM model with different preprocessing methods in Figure 5. We observe that both of the curves align the learned and actual values well; hence, it is our conjecture that the LightGBM model with both of the preprocessing methods would perform well in prediction.

We tested the LightGBM model with different preprocessing methods and plotted their corresponding prediction curves in Figure 6, which shows that the predicted values of the LightGBM model with both of the preprocessing methods follow the trend closely with high accuracy compared with the ground truths.

### 4.3. Comparison of Learning Time

We measured the learning time for each model as shown in Table 6. We observe that the LightGBM learning algorithm learns significantly faster than the LSTM learning algorithm and consumes far fewer system resources.

### 4.4. Shapley Value of the LightGBM Learning Model

We conducted further analysis to understand what role these features play in the prediction process. Based on game theory, we computed the contribution of each feature to the score using SHAP (Shapley value) [19], as shown in Figure 7. These results show that ground humidity, 10 m temperature, and rainfall have the largest effects, and the other features also contribute to the prediction in some degree.

### 4.5. Gamma Exposure Rate Prediction

Combining our research, we can perform the gamma exposure rate prediction, given the gamma exposure rate and weather data. If there is a missing data item, we copy the data item from its preceding time step. The entire dataset is standardized. If there are weather data features whose absolute value of the correlation coefficient is less than 0.01, they are removed, and a new dataset is created. If there are weather data features with *p*-value greater than 0.05 in the new dataset, they are also removed to form a new dataset.

The resulting dataset is divided into training data, validation data, and test data. Then, the model is trained using the training data and the performance of the trained model is evaluated using the validation data. The above process is repeated when new data arrives. The flowchart of this prediction process is provided in Figure 8.

## 5. Discussion and Conclusions

In this work, we hypothesized that there exists a certain relationship between the gamma exposure rate and weather data, analyzed the correlation between them, and proposed two machine learning models, LSTM and LightGBM, to predict the gamma exposure rate using various weather data.

In fact, previous studies have shown that there exists a high correlation between the gamma exposure rate and rainfall data. Our study confirms that other environmental parameters such as humidity and temperature also have a significant effect.

Data preprocessing is an important step to get the data ready for model training. We investigated two methods for data preprocessing, i.e., normalization and standardization. Our study shows that if normalization is used for preprocessing data with small deviations, it tends to converge to the average value. Standardization preprocessing leads to larger learning errors but yields better learning results. Standardization is considered more suitable for preprocessing data with small deviations.

The execution time measured in the experiments shows that LightGBM runs much faster in training and consumes far fewer system resources than LSTM. It seems that real-time analysis and prediction are possible with LightGBM running on a single-board computer (such as Jetson Nano, Coral Dev Board, Raspberry Pi, etc.) without the need to transmit data collected from the AWS and ERMS to a remote high-performance server.

The most significant finding of our research is that the gamma exposure rate can be predicted accurately by learning various weather data using the LightGBM learning algorithm. The trained LightGBM model has great potential to help us determine if the increase in the gamma exposure rate is due to a change in the weather or indeed an actual gamma ray. Our approach can also help autonomous vehicles to choose a safe route.

The runtime performance of LightGBM paves a way to realizing edge intelligence through edge computing using a single-board computer. It is in our interest to conduct real-time machine learning-based diagnosis to determine the root cause of a variation in the gamma exposure rate.

## Figures and Tables

**Figure 1 sensors-22-07062-f001:**
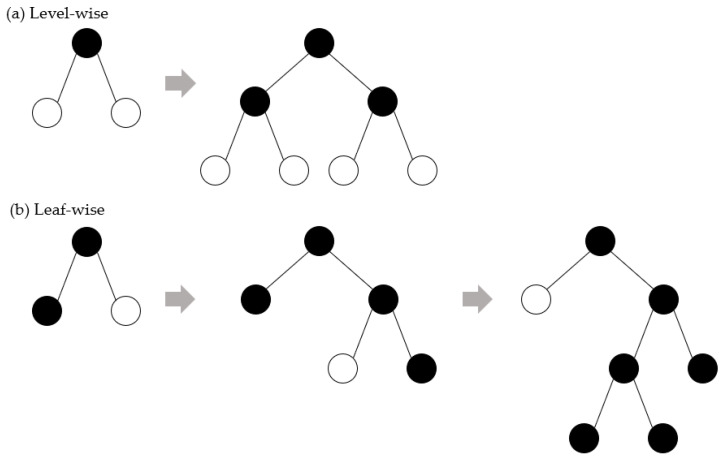
(**a**) Level-wise tree growth; (**b**) leaf-wise tree growth.

**Figure 2 sensors-22-07062-f002:**
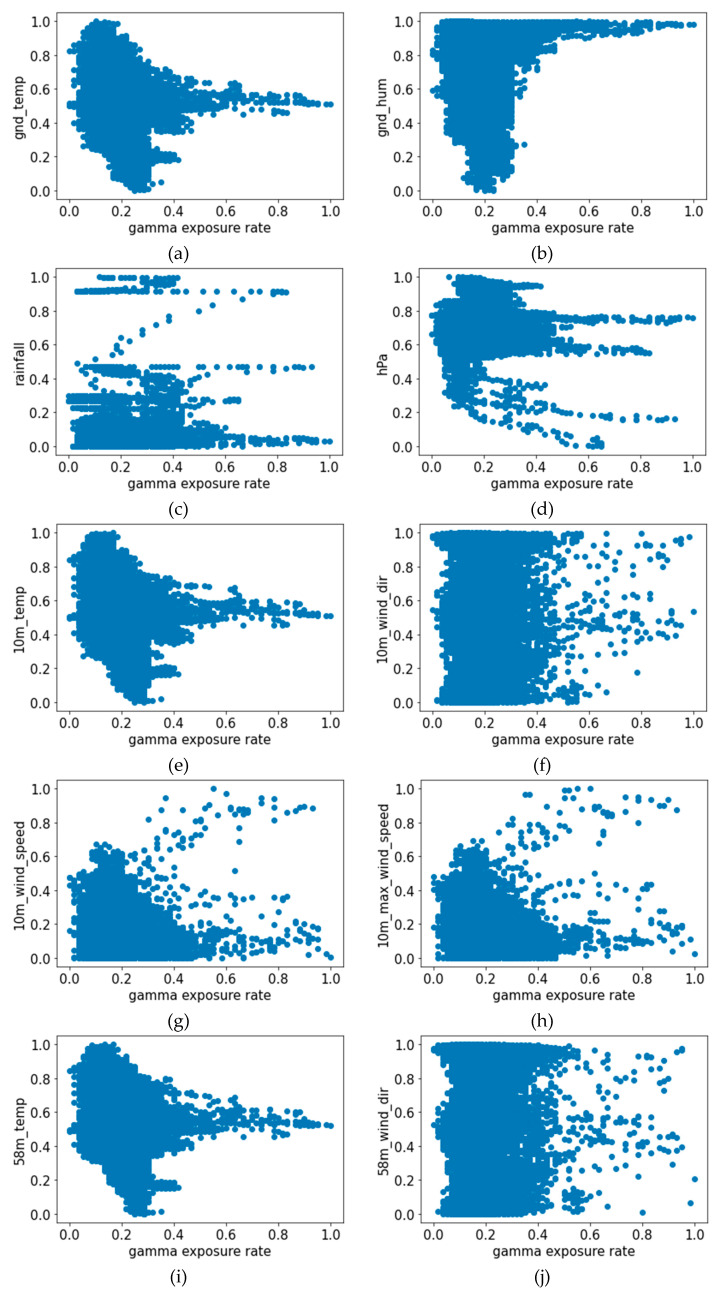
Scatter plot of gamma exposure rate in response to (**a**) ground temperature; (**b**) ground humidity; (**c**) rainfall; (**d**) atmospheric pressure; (**e**) 10 m tower temperature; (**f**) 10 m tower wind direction; (**g**) 10 m tower wind speed; (**h**) 10 m tower maximum wind; (**i**) 58 m tower temperature; (**j**) 58 m tower wind direction; (**k**) 58 m tower wind speed; and (**l**) 58 m tower maximum wind.

**Figure 3 sensors-22-07062-f003:**
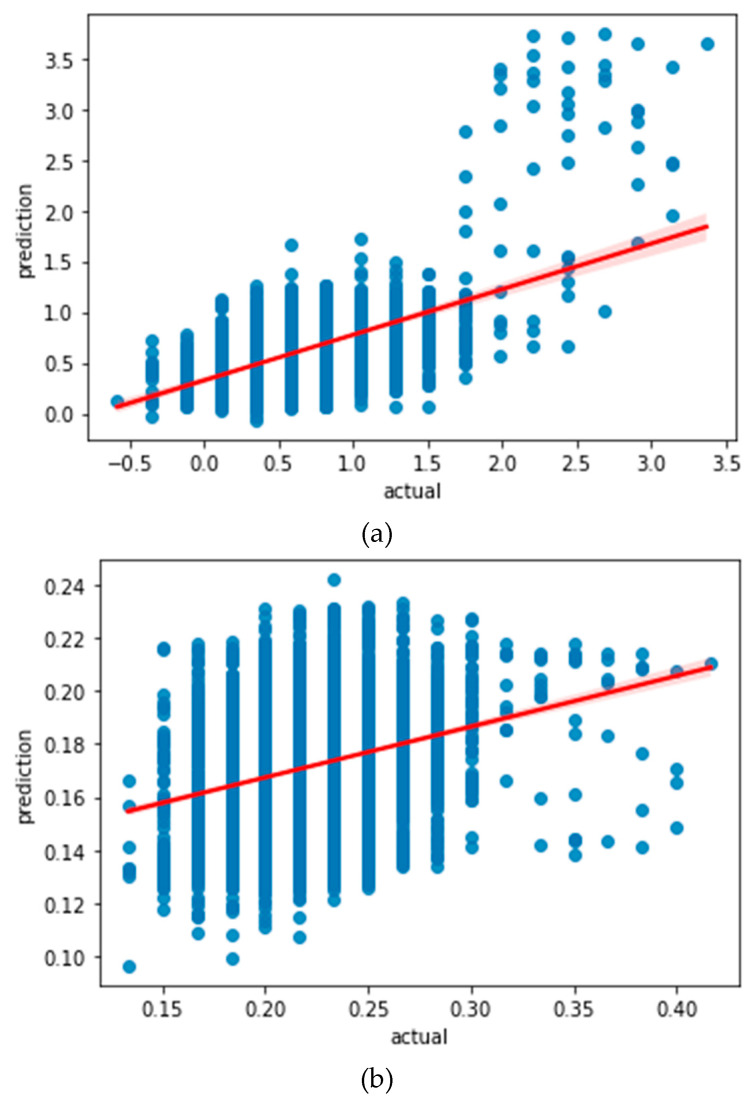
(**a**) Regression curve of LSTM learning model with standardization preprocessing; (**b**) regression curve of LSTM learning model with normalization preprocessing.

**Figure 4 sensors-22-07062-f004:**
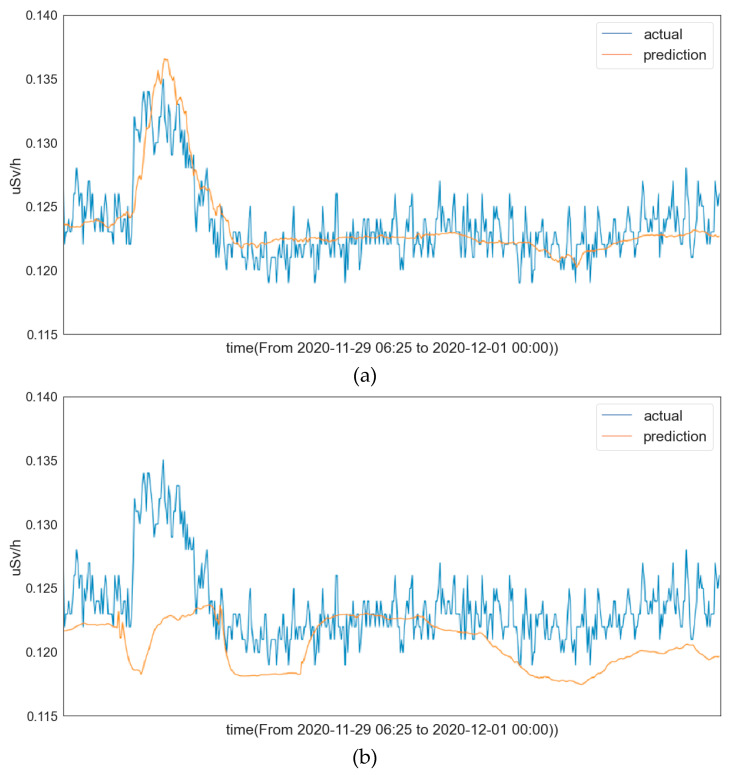
(**a**) Prediction curve of LSTM learning model with standardization preprocessing; (**b**) prediction curve of LSTM learning model with normalization preprocessing.

**Figure 5 sensors-22-07062-f005:**
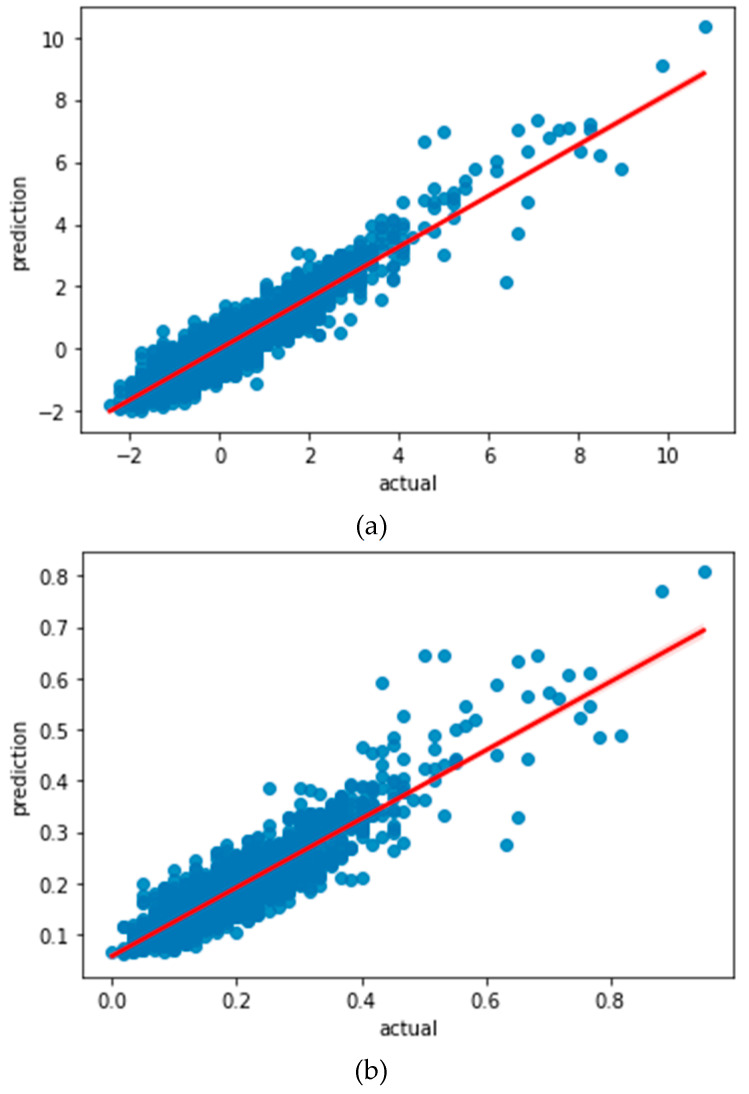
(**a**) Regression curve of the LightGBM learning model with standardization preprocessing; (**b**) regression curve of the LightGBM learning model with normalization preprocessing.

**Figure 6 sensors-22-07062-f006:**
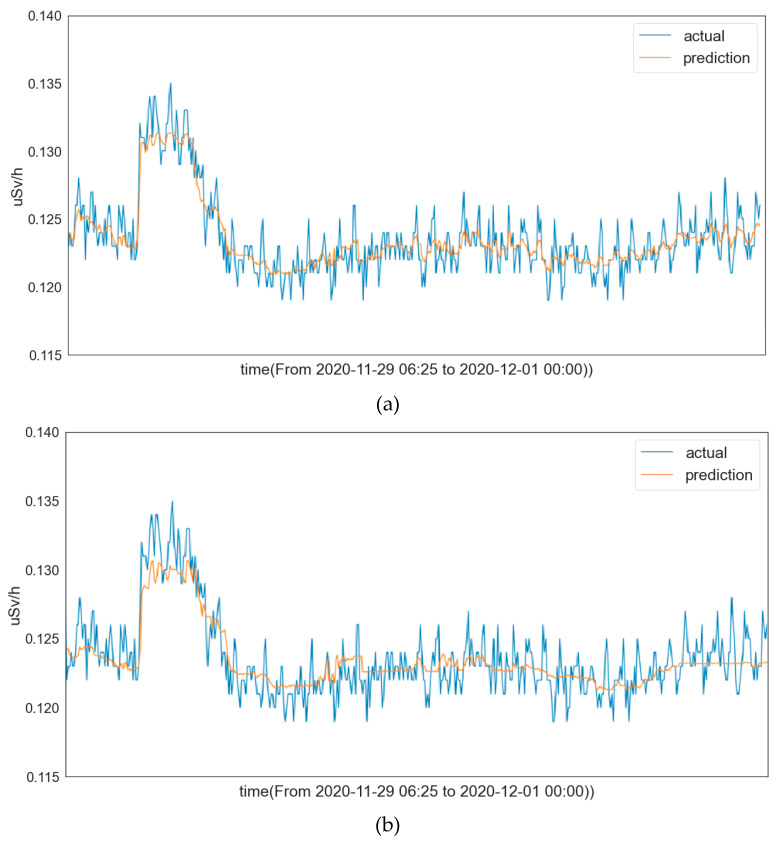
(**a**) Prediction curve of the LightGBM learning model with standardization preprocessing; (**b**) prediction curve of the LightGBM learning model with normalization preprocessing.

**Figure 7 sensors-22-07062-f007:**
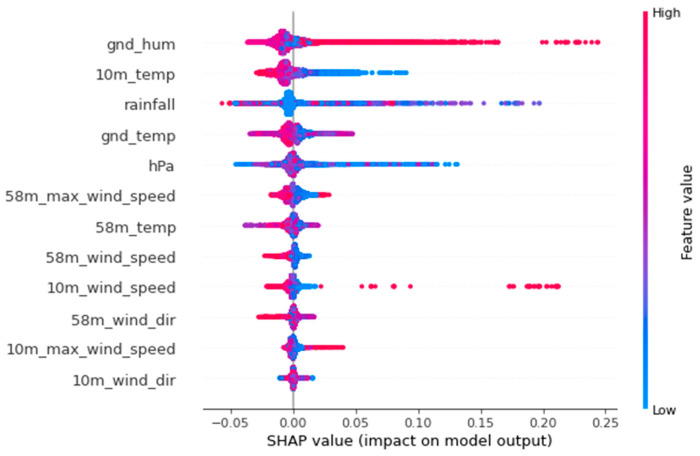
The SHAP value of the LightGBM learning model.

**Figure 8 sensors-22-07062-f008:**
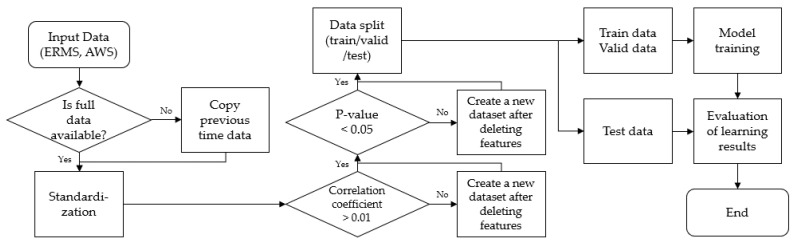
The flowchart of gamma exposure rate prediction.

**Table 1 sensors-22-07062-t001:** Environmental parameters in the experimental dataset.

Data	Unit
Ground Temperature	°C
Ground Humidity	%
Rainfall	mm
Atmospheric Pressure	hPa
10 m Tower Temperature	°C
10 m Tower Wind Direction	Degree
10 m Tower Wind Speed	m/s
10 m Tower Maximum Wind Speed	m/s
58 m Tower Temperature	°C
58 m Tower Wind Direction	Degree
58 m Tower Wind Speed	m/s
58 m Tower Maximum Wind Speed	m/s
Exposure Dose Rate	uSv/h

**Table 2 sensors-22-07062-t002:** Hardware specifications in the experiments.

Type	Specification
CPU	Ryzen 7 5800X 8-Core Processor 3.80 GHz
RAM	DDR4 PC4-25600 16 GB
VGA	GeForce RTX 3080 10 GB
Storage	M.2 NVMe PCIe 3.0 512 GB SSD
Operating System	Windows 11 Pro

**Table 3 sensors-22-07062-t003:** Correlation coefficient and *p*-value between weather data and gamma exposure rate.

Weather Data	Correlation Coefficient	*p*-Value
Ground Temperature	−0.2880	0
Ground Humidity	0.0122	0.0215
Rainfall	0.0663	$7.6658\times 10^{-36}$
Atmospheric Pressure	0.0537	$5.1104\times 10^{-24}$
10 m Temperature	−0.2835	0
10 m Wind Direction	0.0506	$1.6025\times 10^{-21}$
10 m Wind Speed	−0.1378	$9.0102\times 10^{-150}$
10 m Max Wind Speed	−0.1349	$1.3640\times 10^{-143}$
58 m Temperature	0.2600	0
58 m Wind Direction	0.0448	$3.1740\times 10^{-17}$
58 m Wind Speed	−0.1315	$2.3393\times 10^{-136}$
58 m Max Wind Speed	−0.1212	$5.3104\times 10^{-116}$

**Table 4 sensors-22-07062-t004:** Errors of the LSTM model with different preprocessing methods.

	MSE	RMSE
LSTM learning model with standardized preprocessing	0.1611	0.4013
LSTM learning model with normalized preprocessing	0.0019	0.0433

**Table 5 sensors-22-07062-t005:** Errors of the LightGBM model with different preprocessing methods.

	MSE	RMSE
LightGBM learning model with standardization preprocessing	0.1478	0.3844
LightGBM learning model with normalization preprocessing	0.0011	0.0337

**Table 6 sensors-22-07062-t006:** Comparison of the learning time of different learning models.

Model with Preprocessing	Learning Time (s)
LSTM learning model with standardization preprocessing	5120.0298 (Epoch: 22)
LSTM learning model with normalization preprocessing	3093.9445 (Epoch: 16)
LightGBM learning model with standardization preprocessing	10.7164 (Round: 5000)
LightGBM learning model with normalization preprocessing	7.9698 (Round: 3318)

## Data Availability

Not applicable.

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
