# Peer review of "On Weather Data-Based Prediction of Gamma Exposure Rates Using Gradient Boosting Learning for Environmental Radiation Monitoring"

_sensors, 2022, doi:10.3390/s22187062_

Round 1
Reviewer 1 Report (Previous Reviewer 1)
Compared with the previous version of the manuscript, this version had made some improvements, but there are still the following problems:
1. The biggest problem is that this manuscript has no innovation in method. This manuscript only experimentally verifies that lightGBM is superior to LSTM. It is suggested that the authors will improve the innovation of this manuscript. For example, the authors further improve the lightGBM, which is proved to be effective in experiments. Perhaps, the authors propose their own feature selection algorithm, which is proved to be effective in experiments.
2. If correlation analysis cannot help feature selection or get some valuable conclusions, then correlation analysis is of little significance.
3. This manuscript still does not provide the super-parameter tuning process and corresponding experimental results of both LSTM and lightGBM.
Author Response
We are grateful to the reviewer 1 for the feedback and constructive comments. The following is our response to the comments of reviewer 1 from the round 2 of review:
Comment 1. The biggest problem is that this manuscript has no innovation in method. This manuscript only experimentally verifies that lightGBM is superior to LSTM. It is suggested that the authors will improve the innovation of this manuscript. For example, the authors further improve the lightGBM, which is proved to be effective in experiments. Perhaps, the authors propose their own feature selection algorithm, which is proved to be effective in experiments.
Following the reviewer’s suggestion, we provided a subsection to describe our approach in more detail. We’d also like to point out that our main goal in this work is to apply relatively mature machine learning algorithms to solve a practical domain problem and present some data-based scientific findings. Hence, we focused the technical parts of our work on the application of such methods and the evaluation of their effectiveness in various settings (for example, in combination with different data preprocessing methods).
Comment 2. If correlation analysis cannot help feature selection or get some valuable conclusions, then correlation analysis is of little significance.
Thank you for the suggestion. According to the previous research results, it was expected that rainfall has the highest correlation and other features have the lower correlation. However, our analysis showed that other features are even more correlated than rainfall, and that all features could have meaningful results. Based on this analysis result, we proposed a feature selection method in our approach. Since data collected at different time periods may not have this correlation, we also added an additional step in our algorithm to remove uncorrelated features.
Comment 3. This manuscript still does not provide the super-parameter tuning process and corresponding experimental results of both LSTM and lightGBM.
Following the reviewer’s suggestion, we described the parameter tuning process and presented the corresponding experimental results.

Reviewer 2 Report (New Reviewer)
The authors suggests to collect various weather and radiation data from the Automatic Weather System (AWS) and the Environmental Radiation Monitoring System (ERMS) and according to the data to conclude whether an augment in the gamma exposure is caused by a change in the weather or actual gamma rays from radioactive materials.
The paper is well written and interesting. However, I have several concerns:
The algorithm of Figure 1 is unclear. It would be better to give a pseudo code to make it clear how the augmentation of the trees is done in each method.
In figure 2 and table 3, there is no data about gamma rays. It will be better to show the gamma rays data so the reader will be able to better observe the differences.
In Erickson, Mitchell D. and Alfred J. Cavallo. "Technical information for long term surveillance and monitoring", Waste Management (WM’02) Conference, February 24-28, 2002, Tucson, AZ, it is written "Such performance is more easily obtained from gamma radiation monitoring equipment; in most cases a requirement for continuous monitoring is simply not feasible or is prohibitively expensive with current or foreseeable technology.". I would encourage the authors to explain how they suggest overcoming this obstacle. In Y. Wiseman, "Autonomous Vehicles", Encyclopedia of Information Science and Technology, Fifth Edition, Vol. 1, Chapter 1, pp. 1-11, 2020. Available on at: https://u.cs.biu.ac.il/~wiseman/Autonomous-Vehicles-Encyclopedia.pdf it is written " empty vehicles will go alone in dangerous environments or in their way to park", so autonomous vehicles can be of help to better monitoring gamma rays. I would encourage the authors to cite this encyclopedia entry and the conference paper and explain further about the possibilities of gamma rays monitoring.
Author Response
We are grateful to the reviewer 2 for the feedback and constructive comments. The following is our response to the comments of reviewer 2 from the round 2 of review:
Comment 1. The algorithm of Figure 1 is unclear. It would be better to give a pseudo code to make it clear how the augmentation of the trees is done in each method.
Thank you for the suggestion. The pseudocode is provided in the LightGBM paper in the reference.
1. According to the reference, one may reduce the size of the dataset as follows without significantly changing the distribution of the original dataset.
- Calculate and align the gradient of each instance
- Select an instance with a large gradient as much as the ratio of a
- Randomly select the ratio of b among the remaining instances and multiply by weight ( (1-a)/b )
2. When learning the tree, the variance gain corresponding to the split point is calculated, and the split is performed at the point with the largest gain as shown in the figure.
Comment 2. In figure 2 and table 3, there is no data about gamma rays. It will be better to show the gamma rays data so the reader will be able to better observe the differences.
Thank you for this comment. Following the suggestion, we have added gamma rays to the x-axis in Figure 2 and the description in Table 3.
Comment 3. In Erickson, Mitchell D. and Alfred J. Cavallo. "Technical information for long term surveillance and monitoring", Waste Management (WM’02) Conference, February 24-28, 2002, Tucson, AZ, it is written "Such performance is more easily obtained from gamma radiation monitoring equipment; in most cases a requirement for continuous monitoring is simply not feasible or is prohibitively expensive with current or foreseeable technology.". I would encourage the authors to explain how they suggest overcoming this obstacle. In Y. Wiseman, "Autonomous Vehicles", Encyclopedia of Information Science and Technology, Fifth Edition, Vol. 1, Chapter 1, pp. 1-11, 2020. Available on at: https://u.cs.biu.ac.il/~wiseman/Autonomous-Vehicles-Encyclopedia.pdf ; it is written " empty vehicles will go alone in dangerous environments or in their way to park", so autonomous vehicles can be of help to better monitoring gamma rays. I would encourage the authors to cite this encyclopedia entry and the conference paper and explain further about the possibilities of gamma rays monitoring.
Thank you for the suggestion. In Korea, real-time gamma-ray monitoring is being conducted nationwide (Reference site: https://iernet.kins.re.kr/). The work in this manuscript also used the data measured at such monitoring points.
Also, providing autonomous vehicles with predictive data on gamma ray increase helps them choose a safe route. An explanation is provided in the introduction and conclusion by referring to the document provided by Reviewer 2.

Round 2
Reviewer 1 Report (Previous Reviewer 1)
The authors had revised the manuscript according to the previous comments and replied to the innovation of this manuscript. I have no new comments.
Author Response
Comment: The authors had revised the manuscript according to the previous comments and replied to the innovation of this manuscript. I have no new comments.
: We are grateful to the reviewer 1 for the feedback and constructive comments. Thanks to reviewer 1, we were able to create an innovation for the manuscript.
Reviewer 2 Report (New Reviewer)
The authors made a decent effort and the paper is certainly publishable so I would recommend accepting the paper.
Author Response
Comment: The authors made a decent effort and the paper is certainly publishable so I would recommend accepting the paper.
: We are grateful to the reviewer 2 for the feedback and constructive comments. Thanks to Reviewer 2, we were able to broaden the field of application of the manuscript.
This manuscript is a resubmission of an earlier submission. The following is a list of the peer review reports and author responses from that submission.
Round 1
Reviewer 1 Report
The manuscript is mainly to verify which of standardization and normalization is more suitable for preprocessing the data in this manuscript, and which of LSTM algorithm and Light GBM algorithm is more suitable for predicting the gamma exposure rate based on the data in this manuscript. Therefore, there is no innovation in the method. Some comments are as follows.
- Abstract is a complete description, which should not be artificially divided into (1) background, (2) method, (3) result and (4) conclusions.
- The correlation analysis in this manuscript is only to verify that the data are relevant and can be used for analysis. If the data has more dimensional features, it is recommended to use correlation analysis to select the appropriate features.
- For the same data and the same prediction algorithm, using standardization and normalization will certainly get different MSE or RMSE. Comparing the size of MSE or RMSE cannot be used to judge the advantages and disadvantages of standardization and standardization.
- It is suggested to add the tuning process of the LSTM algorithm and Light GBM algorithm, which proves that the parameters of the LSTM algorithm and Light GBM algorithm in this manuscript are optimal.
Reviewer 2 Report
This manuscript developed environmental radiation monitoring system to predict gamma exposure rate using weather data and gradient boosting framework-based learning algorithm. The proposed method was compared to LSTM in terms of performance evaluation, which demonstrated the superiority. The outcome of this research provide a potential solution to the cause of increase in the gamma exposure rate. Overall, the topic of this research is interesting, and the manuscript was well organised and written. I suggest that it can be considered to be accepted for publication in Sensors, if the authors can well address the following comments.
- The contribution and innovation of the manuscript should be clarified clearly in abstract and introduction.
- Broaden and update the literature review to better connect to the current effort in the field of gamma exposure rate prediction based on machine learning.
- How did the authors select hyperparameters of proposed algorithm to achieve the optimal performance?
- Figure 6 shows a sensitivity analysis of model inputs. If some inputs are ignored to diminish the input dimension for simplification of model, can the model performance be affected?
- Some figures are not clear. Please revise and increase font size.
- More future research should be included in conclusion part.